# Affective–Sexual Behaviors in Youth: Analysis of a Public Health Survey in the School Setting

**DOI:** 10.3390/healthcare12171762

**Published:** 2024-09-04

**Authors:** José Antonio Zafra-Agea, Estel·la Ramírez-Baraldes, Cristina García-Salido, Daniel García-Gutiérrez, Mireia Vilafranca-Cartagena

**Affiliations:** 1Department of Nursing, Faculty of Health Sciences at Manresa, University of Vic–Central University of Catalonia, 08500 Vic, Spain; cgarcia@umanresa.cat (C.G.-S.); mvilafranca@umanresa.cat (M.V.-C.); 2Department of Nursing, Faculty of Nursing, Physiotherapy, and Podiatry, University of Seville, 41004 Seville, Spain; 3Intensive Care Unit, Althaia University Health Network, 08243 Manresa, Spain; 4Research Group on Simulation and Transformative Innovation (GRIST), Institute of Research and Innovation in Life and Health Sciences of Central Catalonia (IRIS-CC), 08500 Vic, Spain; 5Research Group on Epidemiology and Public Health in the Context of Digital Health (Epi4Health), Institute of Research and Innovation in Life and Health Sciences of Central Catalonia (IRIS-CC), 08500 Vic, Spain; 6Internal Medicine, Althaia University Health Network, 08243 Manresa, Spain

**Keywords:** adolescents, sexuality, sexual health, pornography consumption, contraceptive knowledge, gender differences

## Abstract

Introduction: Adolescence is a critical period for the development of affectivity and sexuality. Adolescents begin to explore their sexual identities, form intimate relationships, and learn to manage their emotions in new and complex contexts. This study aims to analyze the prevalence of habits and behaviors related to the affective–sexual health of adolescents in the fourth year of ESO, identifying risk factors, comparing their behaviors and risks, monitoring these behaviors, analyzing their pornography consumption, and evaluating the use of the internet as a source of sexual information. Method: Descriptive cross-sectional study using questionnaires. Participants are fourth-year ESO students from a school in the Baix Llobregat province (Catalonia), aged between 14 and 18 years. Descriptive and bivariate statistical analyses are conducted using the chi-square test and *p*-value calculations with the R Project software. Frequency and percentage analyses are also used to describe the health behaviors of the participants. Results: The study reveals that girls have better knowledge of the contraceptive pill and male condom than boys. Sexual initiation and condom use management vary between the genders, with girls being more capable of persuading their partners to use them. Pornography consumption also shows significant differences in terms of the age of initiation and frequency between boys and girls. Conclusions: This study on adolescent affectivity and sexuality reveals knowledge about contraceptives, early sexual initiation, and high pornography consumption, highlighting the need for early and diverse affective–sexual education, youth empowerment, and media misinformation management to promote safer and healthier behaviors within municipal public health.

## 1. Introduction

Adolescence, as a transitional phase between childhood and adulthood [1], is characterized by significant physical, emotional, social, and cognitive changes, leading to ambivalences and contradictions in the process of achieving personal and social equilibrium [2]. This period is crucial for individual development. During this stage, there is a notable shift in emotions and in the ways in which individuals relate to others. These profound changes affect both self-perception and interactions with the social environment, highlighting adolescence as a time of intense transformation. It is a decisive period for the adoption and consolidation of healthy lifestyles, as well as a time for experimentation and the implementation of health-related behaviors, broadly encompassing all factors that contribute to individual well-being and biopsychosocial development. Schools play a crucial role in promoting the health and safety of young people [3].

Recent research highlights that current interventions and policies often focus on specific aspects of adolescent health, such as mental health or substance abuse, but may not fully consider the multifaceted nature of affective–sexual development [4,5]. This study aims to address this gap by providing a comprehensive analysis of adolescents’ affective–sexual health and contributing to the development of more holistic public health strategies. For clarity, “affective–sexual health” in this study refers to the well-being associated with emotional and sexual development during adolescence, including self-perception, relationship dynamics, and sexual behaviors. “Risk behaviors” are defined as actions that potentially jeopardize one’s physical or emotional health, such as unprotected sexual activity or substance abuse [6].

Adolescence is also a critical period for the development of affectivity and sexuality. Adolescents begin to explore their sexual identities, form intimate relationships, and learn to manage their emotions in new and complex contexts [7,8,9]. The ways in which adolescents handle these experiences can have significant implications for their emotional well-being and long-term health. Understanding and accepting one’s sexuality, as well as forming healthy affective relationships, are essential components of this developmental process [10].

This study is grounded in Erikson and Bronfenbrenner’s theories of adolescent development, which explain how adolescents navigate their affective and sexual development within their social environments [11,12]. Additionally, a salutogenic perspective is incorporated to understand how factors and resources in the environment promote well-being and the ability to cope with challenges [13]. This theoretical framework enriches the interpretation of the findings by situating them within the broader context of development and overall health.

Furthermore, this developmental stage is marked by experimentation and risk-taking, which can lead to potentially harmful behaviors, such as substance abuse, unprotected sexual practices, and other actions that may jeopardize both physical and mental health. According to the Spanish Ministry of Health, 25% of adolescents aged 14 to 18 have engaged in sexual intercourse, and approximately 20% of these individuals do not use contraceptives regularly. This increases the risks of unintended pregnancies and sexually transmitted infections (STIs), including human immunodeficiency virus (HIV) and chlamydia. Additionally, the incidence of STIs among Spanish youth has been rising, with cases of gonorrhea and syphilis doubling over the past decade [14].

The consumption of pornography and the use of the internet to seek sexual information are relevant issues during adolescence [15]. The accessibility of the internet has facilitated access to a vast amount of information, including sexual information, which can influence the development of sexual attitudes and behaviors [16]. A study by the Youth Observatory in Spain reveals that 70% of adolescents access online pornographic content before the age of 18, and this consumption can impact their sexual expectations and behaviors, as well as increasing the risk of addiction and mental health issues [17].

Conducting periodic surveys related to adolescents’ health habits can serve as a valuable tool for the monitoring of behaviors, with the aim of establishing future interventions for health promotion and the prevention of maladaptive behaviors. It is noteworthy that adolescence is considered a crucial stage for research studies, as it is seen as a “sentinel agent” for future consolidated and maintained behaviors in adulthood [18].

This study also seeks to address a gap in the literature by integrating various dimensions of affective–sexual health into a single comprehensive assessment, thus providing a more holistic understanding of adolescent health. This integration aims to support the development of targeted interventions that address multiple aspects of adolescent well-being simultaneously.

Due to changes in policies related to the promotion and prevention of affective–sexual health habits in primary care, this study aims to provide a comprehensive view of adolescents’ affective–sexual health, with the goal of offering practical recommendations to improve sexual education and public health policies. Therefore, the objective of this study is to analyze the prevalence of affective–sexual-health-related habits and behaviors among adolescents enrolled in the fourth year of secondary education (fourth ESO), identifying risk factors, comparing their behaviors and risks, monitoring these behaviors, analyzing their pornography consumption, and evaluating their use of the internet as a source of sexual information.

This local study, supplementary to the Local Health Indicator Reports, is designed to support the formulation, implementation, evaluation, and prevention of local health policies, with a particular emphasis on health promotion and education [19]. At the municipal level, this study is essential in guiding and optimizing local health policies, as it provides key data that allow for the adaptation and application of specific strategies based on the unique needs of each community. Its use ensures a more accurate and effective response to local health challenges, facilitating informed decision-making and the continuous improvement of public health interventions, as well as the prevention of health issues before they arise.

## 2. Methodology

### 2.1. Study Design

This study employed a cross-sectional descriptive design with a quantitative approach, conducted in a municipality in the province of Barcelona during the 2023–2024 academic year. The aim of the study was to analyze the prevalence of affective–sexual health-related habits and behaviors among adolescents enrolled in the 4th year of secondary education (4th ESO).

### 2.2. Study Population

Study Population: The study included 120 students from the 4th year of ESO at the selected school. Participants completed an online questionnaire individually and anonymously during class hours, following the acquisition of informed consent from their legal guardians. Participation in the study was voluntary, and data were collected confidentially and used only in aggregate form.

Response Rate: Out of the 120 students invited, 86.3% completed the survey. This percentage represents the effective response rate for the sample.

Differences Between Respondents and Non-Respondents: The initial analysis did not include an assessment of the differences between students who responded and those who did not respond to the survey. Due to the anonymous and confidential nature of the survey, as well as the data collection process, a detailed analysis to identify significant differences between these groups was not conducted.

### 2.3. Ethical Procedures

This study was conducted in accordance with the ethical guidelines set forth in the Declaration of Helsinki. Informed consent was obtained from the legal guardians to ensure the voluntary, anonymous, and confidential participation of the students. The study was approved by the ethics committee of the educational institution and the public health service of the municipal government, with approval number SP:23/002 (19 September 2023). Each participant provided written informed consent.

### 2.4. Data Collection Instruments

An online questionnaire consisting of 76 validated questions, categorized into various health-related domains, was used for this study. The questionnaire covered topics such as sexuality, the use of contraceptive methods, the frequency of sexual activity, the use of emergency contraception, the incidence of pregnancies, condom use, pornography consumption, and the use of the internet as a source of sexual information. The objective of the study was to analyze the prevalence of affective–sexual-health-related habits and behaviors among adolescents enrolled in the 4th year of secondary education (4th ESO), identify risk factors, compare their behaviors and risks, monitor these behaviors, and evaluate the impact of pornography consumption and online information on their sexual health.

### 2.5. Variable Recategorization

Some independent variables were recategorized by grouping the responses to ensure that each category contained at least 10% of the cases.

### 2.6. Data Collection Procedure

The survey was distributed electronically to students during class hours. Data collection took place between November 2023 and February 2024.

### 2.7. Statistical Analysis

Descriptive statistics, including frequencies, percentages, means, and standard deviations (SD), were calculated. Pearson’s chi-square test was employed to compare groups and to assess the associations between demographic characteristics, knowledge, attitudes, and practices related to sexuality. A *p*-value of <0.05 was considered statistically significant. All analyses were performed using the R statistical software v. 4.4.1 (R Project for Statistical Computing).

## 3. Results

### Sample Description

The sample comprised 120 students from the fourth year of secondary education (fourth ESO) at a public high school in a town in the province of Barcelona, aged between 14 and 18 years. The average age was 15.2 years, with a gender distribution of 57.3% female and 42.7% male. The majority of the students were native (71.8%), with 3.2% being first-generation immigrants, 16.9% second-generation immigrants, and 8.1% who did not specify their background.

Regarding the family structure, 71% had a biparental family, 24.2% had a single-parent family, and 3.2% had a restructured family. The socioeconomic status was classified as high for 63.7%, medium for 29.8%, and low for 6.5%.

The educational levels of the parents showed that most had secondary education (39.1%) or university degrees (28.2%). A small percentage had no formal education (5.6%) or primary education (8.5%), and 18.6% did not specify or were unsure.

Table 1 displays the knowledge of contraceptive methods among the fourth-year students. It shows that knowledge of the contraceptive pill and male condoms is significantly higher among females (98.6%) compared to males (90.2%), with *p*-values of 0.013 in both cases. Conversely, knowledge of female condoms is similar between males (90.2%) and females (91.3%), with no significant difference (*p* = 1.000). Additionally, 78.4% of males are aware of diaphragms, compared to 73.9% of females, with no statistically significant difference (*p* = 0.619). Finally, knowledge of intrauterine devices and spermicides is also greater among males, although these differences are not statistically significant (*p* = 0.317 for intrauterine devices and *p* = 0.066 for spermicides).

Table 2 presents data on penetrative sexual intercourse. The percentage of males who report having engaged in penetrative sexual intercourse is 23.5%, compared to 27.5% of females, with no significant difference (*p* = 0.122). In contrast, a higher percentage of males (68.6%) report never having had penetrative sexual intercourse, compared to 63.8% of females, with a significant difference (*p* = 0.041). Knowledge of non-penetrative sexual intercourse is similar between males (7.8%) and females (7.2%), with a significant difference (*p* = 0.008). Regarding the frequency of penetrative sexual intercourse, 16.7% of males report engaging in it several times a week, compared to 31.6% of females, although this difference is not statistically significant (*p* = 0.438). The frequency of “occasional” participation is similar between males (33.3%) and females (31.6%), with no significant difference (*p* = 0.077). The category of “rarely” also shows similar percentages between males (25%) and females (31.6%), with no significant difference (*p* = 0.334). Notably, 25% of males and only 5.3% of females report having had sexual intercourse only once, but this is not statistically significant (*p* = 0.930).

The analysis in Table 3, which examines adolescents’ ability to manage condoms, reveals that 69.6% of females feel completely capable of persuading their partner to use condoms, compared to 51.0% of males, with a significant difference (*p* = 0.048). Additionally, 21.6% of males feel completely incapable of always carrying condoms, whereas only 8.7% of females feel this way, showing a difference approaching significance (*p* = 0.058). Regarding the ability to refuse to engage in unprotected sex, 39.2% of males and 49.3% of females feel completely capable of doing so, with no significant difference (*p* = 0.277). Overall, most differences between males and females regarding condom use are not statistically significant, except for the ability to persuade a partner to use condoms, where females perceive themselves as more capable (*p* = 0.048). Some differences, such as the ability to always carry condoms, approach the significance threshold but do not reach it (*p* = 0.058), suggesting that, in general, both males and females have similar perceptions of their abilities in these areas, with a few exceptions.

Overall, 39.5% of adolescents consider that condoms are the most reliable contraceptive method, while 50% believe that they should always be used. Additionally, 62.9% think that using condoms effectively prevents the transmission of sexually transmitted infections (STIs), and 48.4% view it as an effective solution to avoid unwanted pregnancies. Although 31.5% of adolescents think that their friends believe that condoms should always be used, only 11.3% feel that condom use reduces sexual pleasure.

Table 4 examines the opinions on condom use among boys and girls. Overall, 39.5% of boys and 44.9% of girls consider condoms to be the most reliable contraceptive method, with no significant differences (*p* = 0.258). Regarding the necessity of always using condoms, 50.0% of boys and 43.5% of girls agree, with no significant differences (*p* = 0.127). Both genders agree that condoms are effective in preventing sexually transmitted infections (62.9%) and unwanted pregnancies (48.4%), with no significant differences in these perceptions (*p* = 1.000). Additionally, 31.5% of both boys and girls believe that their friends also think that condoms should always be used, with no significant differences (*p* = 1.000). Finally, 48.4% of both boys and girls believe that condoms do not significantly reduce sexual pleasure, with no significant differences between the genders (*p* = 1.000).

Table 5 examines pornography consumption among boys and girls. The results indicate that 60.8% of boys and 53.6% of girls have ever viewed pornography, with no significant differences (*p* = 0.710). However, significant differences are observed in the age at first exposure: 12.9% of boys began viewing it at age 10 or younger, whereas none of the girls started at this age (*p* < 0.001). Additionally, 45.9% of girls began at age 13, compared to only 6.5% of boys (*p* < 0.001). Regarding the frequency of consumption, 12.9% of boys report excessive consumption, while no girls report the same (*p* < 0.001). Boys also tend to consume pornography habitually, with 22.6%, in contrast to 8.1% of girls, a difference approaching significance (*p* = 0.069). Furthermore, 58.1% of boys consume it occasionally, similar to 45.9% of girls, with no significant differences (*p* = 0.958). Additionally, 45.9% of girls have viewed pornography only once, whereas only 6.5% of boys have done so (*p* < 0.001). Regarding the motivations for the consumption of pornography, 71% of boys do so to experience sexual desire, compared to 37.8% of girls, a difference nearing significance (*p* = 0.063). Meanwhile, 25.8% of boys consume it out of curiosity, while 43.2% of girls do so for the same reason, with no significant differences (*p* = 0.693). Finally, 3.2% of boys consume pornography to learn, compared to 10.8% of girls, with no significant differences (*p* = 0.086).

## 4. Discussion

This study examines the incidence of sexual activity, contraceptive use, and perceptions related to sexuality among adolescents. The findings provide a comprehensive view of the sexual behaviors and attitudes in this population, and their comparative analysis with previous studies enriches the interpretation of these results. We conclude with evidence-based recommendations to enhance sexual education and prevention strategies, aiming to promote healthy and safe sexual development among adolescents.

Knowledge of Contraceptive Methods. The results indicate a high level of knowledge among adolescents about various contraceptive methods. The contraceptive pill and condoms are the most well-known methods, with notable familiarity among both boys (90.2%) and girls (98.6%). In contrast, spermicides are less known (boys: 58.2%, girls: 43.5%). Our findings regarding the high awareness of condoms are consistent with previous studies showing widespread familiarity with these methods among adolescents [20,21]. However, the lower knowledge of less conventional methods, such as spermicides, aligns with the existing literature, suggesting the insufficient promotion of these methods [22]. Research has highlighted that the promotion of and education about less common contraceptive methods, like spermicides, have been inadequate, contributing to lower awareness among youth [23]. These results underscore the importance of improving the education and promotion of a range of contraceptive methods to ensure that adolescents have access to comprehensive and accurate information about all available options.

Sexual Activity. Our results show that the median age for the first instance of sexual intercourse is 14.5 years. Approximately 23.5% of boys and 27.5% of girls have engaged in penetrative sex, while 7.8% of boys and 7.2% of girls have engaged in non-penetrative sex. About 68.6% of boys and 63.8% of girls have not had sexual intercourse.

Previous studies also observing sexual initiation found the average age of the first sexual experience in Spanish adolescents to be between 16 and 18 years [24,25]. Another study noted that 12.8% of adolescents reported having sexual intercourse before age 14 [26]. Additionally, prior research suggests that the age of sexual initiation has remained relatively stable in recent decades at around age 13 [21,27]. De Graaf et al. (2024) reported a decrease in early sexual initiation among 15 year olds in 33 European countries, highlighting the influence of gender norms and gender inequality on these behaviors.

Evidence indicates that adolescents who initiate sexual activity before age 15 tend to engage in a greater variety of sexual practices and exhibit less consistent condom use [24]. These data reflect a concerning trend toward early sexual initiation, potentially influenced by factors such as peer pressure, the sexual education received, and media influences. Additionally, [28] provides a systematic review of the risk factors associated with early sexual initiation, emphasizing the impact of individual, familial, and social factors [16,29].

Contraceptive Use and Pregnancy. About 25.0% of boys and 10.5% of girls do not use contraceptive methods during sexual intercourse. Approximately 8.3% of boys use unsafe methods, while no girls report using these methods. Moreover, 94.7% of girls use barrier methods and contraceptives, compared to 75.0% of boys.

The higher use of contraceptive methods among girls is consistent with previous studies indicating that women tend to adopt more contraceptive methods than men. The discrepancy in barrier method use between the genders also reflects the barriers faced by adolescent males [30,31]. About 43.1% of boys and 49.3% of girls feel capable of rejecting sex without a condom. A systematic cohort study found that mental health issues and beliefs and attitudes are significant risk factors for early sexual initiation. Substance use, such as tobacco and alcohol, also showed a high correlation with this behavior [28]. Regarding opinions about condoms, 39.5% of boys and 62.9% of girls consider them the safest contraceptive method. A study found that 75.9% of adolescents were aware of risky sexual behaviors [32].

Nevertheless, in terms of the actual use of contraceptive methods among those who are sexually active, 75% of adolescents reported using barrier methods and combined contraceptives, while 8.3% used only contraceptive methods. A concerning data point is that 16.7% of the respondents did not use any form of protection during sexual intercourse, indicating a significant gap between knowledge and practice [26]. One justification could be that users of long-acting reversible contraceptives (LARC) are less likely to use condoms, which could increase the risk of STIs [33,34]. Conversely, the observed tendency among girls to use emergency contraception suggests greater awareness or access to this method among them. The low reported incidence of pregnancies (5.3% among girls) indicates that, despite challenges in using protection, most young people are avoiding unintended pregnancies. These results provide crucial information for the development of more effective sexual education programs, highlighting areas needing increased attention and resources.

Perceived Ability to Purchase and Use Condoms. Among boys, 35.3% consider themselves fully capable of purchasing condoms, while 44.9% of girls feel completely capable. Regarding the ability to carry condoms at all times, 29.4% of boys and 43.5% of girls feel fully capable. Regarding the ability to convince a partner to use condoms, 51.0% of boys and 69.6% of girls feel fully capable. Contraceptive use is high, with 75% of boys and 94.7% of girls using combined barrier and contraceptive methods. This aligns with previous research on sexual practices among Spanish adolescents, noting that 82% use some form of contraceptive, with condoms being the most common [25]. The difference in the perceived ability to manage condoms between boys and girls may reflect differences in education and social expectations related to contraceptive use. Girls tend to feel more capable in various aspects related to condom use, which may be linked to the greater social pressure to assume responsibilities in sexual protection [35]. Furthermore, evidence indicates that young women tend to negotiate contraceptive use more effectively [36,37,38]. These studies highlight the need to address social and educational expectations in sexual education programs to reduce the gender disparities in the perception and use of contraceptive methods.

Sources of Information about Sexuality. Family is a significant source of sexual education for boys (58.0%), while school is the primary source for both genders, being more influential for boys (76.5%). Friends and social media also play significant roles, with boys using social media more (41.2%) compared to girls (30.4%). Pornography is mentioned by 31.4% of boys and only 2.9% of girls. The internet is becoming increasingly important as a source of information and sexual education, consistent with evidence, especially in the later years of adolescence and among boys [26]. It is important to emphasize the need for comprehensive sexual education from an early age to address the myths and misinformation adolescents may have about sexuality [39]. Our findings align with Rivera et al.’s (2021) integrative review, which also found that educational interventions can be effective in preventing risky behaviors in adolescents, highlighting the importance of sexual education as a tool to prevent sexual violence and promote responsible sexual behaviors.

Our study’s findings on sexual education among adolescents are consistent with existing evidence on the effectiveness of comprehensive sexual education (CSE). According to a systematic review of three decades of research, CSE in schools is effective not only in preventing pregnancy and sexually transmitted infections but also in promoting the appreciation of sexual diversity, preventing partner violence, and improving socio-emotional learning and media literacy [40]. These findings reinforce the importance of implementing sexual education programs that address a broad range of topics and adopt inclusive and affirmative approaches to human sexuality. It is essential to increase the quantity and quality of sexual education in schools, including practical training on contraceptive methods and the importance of protection. Developing and promoting reliable online resources and providing training and resources for parents to offer accurate and open information about sexuality at home is crucial.

Relationship with Pornography. About 60.8% of boys and 53.6% of girls have viewed pornography, with early exposure at age 10 or younger being more common among boys (25.8%) compared to girls (32.4%). Frequent consumption is reported by 22.6% of girls and 8.1% of boys, while 12.9% of boys report excessive consumption.

The main reasons cited for consuming pornography include curiosity (43.2%) and sexual desire (37.8%), with less interest in learning about sexuality (10.8%) and a smaller influence from friends (8.1%). Approximately two thirds of adolescents have had their first experience with pornography, and 52.2% use it at least once a week. These findings suggest that pornography serves as a significant source of sexual information and stimulation for many adolescents [41]. According to evidence [42], adolescents tend to have their first experience with pornography during early adolescence, with an average age at first use of around 12 years [42].

The differences in the frequency and reasons for pornography consumption between the genders may reflect their different attitudes and behaviors toward sexuality. The habitual and excessive consumption of pornography among both boys and girls suggests a potential dependency on this type of content for sexual information. These patterns are consistent with previous studies showing greater pornography exposure among male adolescents, indicating that men often consume more pornography and have earlier exposure compared to women [33,41,42,43,44,45,46,47]. This study highlights the significant differences in the exposure to and consumption of pornography between boys and girls. Boys are typically exposed at an earlier age and consume pornography more frequently for sexual satisfaction. In contrast, girls are more likely to encounter pornography accidentally and consume it out of curiosity. These findings underscore the need for comprehensive sexual education that addresses gender differences and promotes a healthy and critical understanding of pornography among youth.

### 4.1. Recommendations

To effectively address the challenges in adolescent sexual and reproductive health, it is crucial to expand the available educational programs. Practical training on the correct use of contraceptive methods and an emphasis on the importance of protection in all sexual relationships are fundamental. This expansion should ensure that adolescents not only have theoretical knowledge but also can apply this knowledge effectively in their daily lives. Additionally, easy and confidential access to contraceptive methods through school and community health services should be ensured. Implementing awareness campaigns to promote the consistent use of contraceptives and educate individuals about the consequences of inconsistent use is essential to bridge the gap between knowledge and practice.

Comprehensive Sexual Education: The study by Goldfarb and Lieberman (2021b) suggests that comprehensive sexual education (CSE) is effective not only in preventing pregnancy and sexually transmitted infections but also in areas such as appreciating sexual diversity, preventing dating violence, and enhancing social/emotional learning [40].

School and Community Interventions: The systematic review by Mason-Jones et al. (2016) emphasizes the need to combine sexual education with incentives to increase its effectiveness. Moreno et al. (2014) suggest that structural and community interventions can improve condom use and knowledge, although they may not necessarily reduce STI transmission [48,49].

Feminist and Inclusive Perspective: The Guía Salut Sexual (2022) highlights the importance of feminist and inclusive sexual education that breaks taboos and promotes diverse identities and sexual orientations [50]. This approach can foster an educational environment that prevents risks and promotes the sexual and emotional well-being of young people.

Multidimensional Approaches: Reis et al. (2023) emphasize the importance of addressing multiple dimensions of risk factors, including individual, familial, social, and environmental aspects, to understand and prevent early sexual initiation among adolescents [28].

### 4.2. Study Limitations

This study provides a detailed view of sexual health among adolescents within a specific population but has several significant methodological limitations. The cross-sectional design used does not allow for the establishment of causal relationships, limiting the ability to infer how certain factors may directly influence sexual behaviors. Additionally, the use of self-reports may introduce social desirability bias and memory errors, as adolescents might not recall accurately or may adjust their responses to align with what they perceive as socially acceptable. The lack of sample representativeness also limits the generalizability of the findings to the broader adolescent population in Spain, suggesting the need for a more diverse and representative sample in future research. To address these limitations, further research is recommended to explore the long-term impact of pornography exposure and the barriers to consistent contraceptive use. Such future research is crucial in developing more effective educational strategies and public health policies that promote healthy and safe sexual behaviors among Spanish adolescents.

## 5. Conclusions

This study titled “Affective–Sexual Behaviors in Youth: Analysis of a Public Health Survey in the School Setting” reveals significant findings that highlight both areas of knowledge and ongoing challenges in adolescent sexual health. A high level of awareness about contraceptive methods such as the pill and condoms was observed, although less common methods like spermicides are less well known. This indicates the need for broader and more diverse education in this area. The average age at sexual initiation is 14.5 years, with a significant proportion of adolescents already sexually active, underscoring the importance of early educational interventions to promote safe sexual practices.

The consumption of pornography is prevalent among adolescents, with early exposure and significant gender differences regarding the reasons for and frequency of consumption. These results emphasize the need to integrate education about pornography and its potential impacts into sexual education programs, promoting a critical and healthy understanding of this content.

Adolescents’ ability to obtain and use condoms varies, with girls reporting greater confidence in managing situations related to contraceptive use. This suggests a need to empower boys in this regard, addressing the gender disparities in sexual education.

Regarding sources of information, family, school, and the media play key roles, although pornography and social media emerge as significant sources, especially among boys. This highlights the importance of providing accurate and accessible information from an early age to counter potential misinformation.

The study has several limitations, such as its cross-sectional design and reliance on self-reports, which may introduce biases. Additionally, the sample is not fully representative of the entire adolescent population in Spain. Future research is recommended to explore the long-term impact of pornography exposure and the barriers to the consistent use of contraceptive methods.

These findings provide a solid foundation for the development of policies and educational programs that address the specific needs of adolescents, promoting comprehensive and safe sexual health. It is crucial that interventions are inclusive and consider gender differences, ensuring that all young people have access to quality sexual education that prepares them to make informed and responsible decisions about their sexual and reproductive health.

## Figures and Tables

**Table 1 healthcare-12-01762-t001:** Contraceptive methods known.

Contraceptive Method	Boys (N = 51)	Girls (N = 69)	*p*-Value
Birth Control Pill	90.2% (n = 46)	98.6% (n = 68)	0.013
Male Condom	90.2% (n = 46)	98.6% (n = 68)	0.013
Female Condom	90.2% (n = 46)	91.3% (n = 63)	1.000
Diaphragm	78.4% (n = 40)	73.9% (n = 51)	0.619
Intrauterine Device (IUDI)	80.4% (n = 41)	72.5% (n = 50)	0.317
Spermicides	58.2% (n = 30)	43.5% (n = 30)	0.066

**Table 2 healthcare-12-01762-t002:** Frequency and penetrative sexual intercourse.

Penetrative Sexual Intercourse	Boys (N = 51)	Girls (N = 69)	*p*-Value
Yes	23.5% (n = 12)	27.5% (n = 19)	0.122
No	68.6% (n = 35)	63.8% (n = 44)	0.041
Yes, without penetration	7.8% (n = 4)	7.2% (n = 5)	0.008
**Total**	**100%**	**100%**	
**Frequency of Penetrative Sexual Intercourse**			
Several times per week	16.7% (n = 8)	31.6% (n = 22)	0.438
Occasionally	33.3% (n = 17)	31.6% (n = 22)	0.077
Rarely	25% (n = 13)	31.6% (n = 22)	0.334
Only once	25% (n = 13)	5.3% (n = 3)	0.930
**Total**	**100%**	**100%**	
**Protection Methods Used**			
None	25% (n = 13)	10.5%(n = 7.2)	0.121
Unsafe methods	8.3% (n = 4)	26.3% (n = 18.1)	0.080
Barrier and contraceptive methods	75% (n = 38)	94.70% (n = 65.3)	0.639
Contraceptive methods only	0% (n = 0)	10.5%(n = 7.2)	0.080
**Total**	**100%**	**100%**	

**Table 3 healthcare-12-01762-t003:** Ability to purchase condoms, keep them on hand, convince a partner to use them, or refuse to engage in unprotected sex.

Ability To:		Boys (N = 51)	Girls (N = 69)	*p*-Value
Purchase Condoms	Completely Capable	35.3% (n = 18)	44.9% (n = 31)	0.258
	Quite Capable	39.2% (n = 20)	31.9% (n = 22)	0.430
	Quite Incapable	11.8% (n = 6)	13.0% (n = 9)	0.849
	Completely Incapable	13.7% (n = 7)	10.1% (n = 7)	0.582
Always Carry Condoms	Completely Capable	29.4% (n = 15)	43.5% (n = 30)	0.127
	Quite Capable	31.4% (n = 16)	21.7% (n = 15)	0.259
	Quite Incapable	15.7% (n = 8)	26.1% (n = 18)	0.168
	Completely Incapable	21.6% (n = 11)	8.7% (n = 6)	0.058
Convince Partner to Use Condoms	Completely Capable	51.0% (n = 26)	69.6% (n = 48)	0.048
	Quite Capable	27.5% (n = 14)	18.8% (n = 13)	0.305
	Completely Incapable	15.7% (n = 8)	8.7% (n = 6)	0.242
Refuse to Engage in Unprotected Sex	Completely Capable	39.2% (n = 20)	49.3% (n = 34)	0.277
	Quite Capable	17.6% (n = 9)	17.4% (n = 12)	0.976
	Quite Capable	17.6% (n = 9)	11.6% (n = 8)	0.424
	Completely Incapable	25.5% (n = 13)	20.3% (n = 14)	0.525

**Table 4 healthcare-12-01762-t004:** Sources of sexual knowledge.

Source of Knowledge	Boys (N = 51)	Girls (N = 69)	*p*-Value
Family	43.1% (n = 22)	58.0% (n = 40)	0.376
School	76.5% (n = 39)	66.7% (n = 46)	0.891
Friends	62.7% (n = 32)	59.4% (n = 41)	0.904
Social media	41.2% (n = 21)	30.4% (n = 21)	0.812
Internet	37.3% (n = 19)	31.9% (n = 22)	0.739
Pornography	31.4% (n = 16)	2.9% (n = 2)	0.013

**Table 5 healthcare-12-01762-t005:** Pornography.

Have You Ever Watched Pornography?	Boys (N = 51)	Girls (N = 69)	*p*-Value
Yes	60.8 (n = 31)	53.6% (n = 37)	0.710
No	35.3% (n = 18)	44.9% (n = 31)	0.498
Not sure/prefer not to say	3.9% (n = 2)	1.4% (n = 1)	0.518
**Total**	**100%**	**100%**	
**At What Age Did You Start Watching Pornography?**			
10 years old or younger	12.9% (n = 6)	0%	<0.001
11	22.6% (n = 12)	8.1% (n = 5)	0.070
12	58.1% (n = 30)	45.9% (n = 32)	0.957
13	6.5% (n = 3)	45.9% (n = 32)	<0.001
14	12.9% (n = 6)	32.4% (n = 22)	0.187
15	3.2% (n = 2)	18.9% (n = 13)	00094
**Total**	**100%**	**100%**	
**Frequency of Pornography Consumption**			
Excessive	12.9% (n = 6)	0% (n = 0)	<0.001
Habitual	22.6% (n = 12)	8.1% (n = 5)	0.069
Occasionally	58.1% (n = 30)	45.9% (n = 32)	0.958
Only once	6.5% (n = 3)	45.9% (n = 32)	<0.001
**Total**	**100%**	**100%**	
**Reasons for Consuming Pornography**			
To learn	3.2% (n = 2)	10.8% (n = 8)	0.086
To experience sexual desire	71% (n = 36)	37.8% (n = 26)	0.063
Out of curiosity	25.8% (n = 13)	43.2% (n = 30)	0.693
Suggested by friends	0% (n = 0)	8.1% (n = 6)	0.280
**Total**	**100%**	**100%**	

## Data Availability

The data presented in this study are available upon request from the corresponding author.

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
