# Peer review of "Affective–Sexual Behaviors in Youth: Analysis of a Public Health Survey in the School Setting"

_healthcare, 2024, doi:10.3390/healthcare12171762_

Round 1

Reviewer 1 Report

Comments and Suggestions for Authors

José Antonio Zafra-Agea et al. submitted to Healthcare an article, dealing with the affective-sexual behaviors in a Spanish school setting.

This manuscript appears well structured, however, it is necessary to clarify some aspects, making appropriate changes.

Please, specify how the questionnaire was validated in terms of intelligibility, during a pilot study, showing the results of the Cronbach's alpha coefficient.

References should be indicated in the text according to the Journal rules (in square brackets).

Comments on the Quality of English Language

Minor editing of English language required.

Author Response

Dear Reviewer 1,

Thank you for your comments on the manuscript. I would like to address your observations as follows:

Comment: This manuscript appears well structured, however, it is necessary to clarify some aspects, making appropriate changes.

Please, specify how the questionnaire was validated in terms of intelligibility, during a pilot study, showing the results of the Cronbach's alpha coefficient.

Reply: The aim of the study is to analyze the prevalence of habits and behaviors related to affective-sexual health among adolescents in 4th year of ESO. Specifically, the study focuses on:

Identification of Risk Factors: Identifying risk factors associated with affective-sexual health in this age group.

Behavior Comparison: Comparing behaviors and associated risks across different groups or time periods.

Behavior Monitoring: Tracking and monitoring changes in these behaviors over time.

Analysis of Pornography Consumption: Examining patterns and implications of pornography consumption among adolescents.

Evaluation of Internet Use: Assessing how adolescents use the internet as a source of sexual information and its impact on their affective-sexual health.

The supplementary questionnaire to the Local Health Indicators Reports is designed to assist in guiding, designing, and evaluating local health policies, with a particular focus on health promotion and education. At the municipal level, this instrument is crucial for directing, designing, and assessing local health policies, providing essential data for the implementation of specific strategies tailored to local needs.

The questionnaire will be provided online, and technical guidance will be offered to schools on how to administer the surveys. After collecting the surveys, the results will be analyzed, and a detailed report will be provided to the municipalities, including a comparison with reference values. Additionally, municipalities that have participated in previous surveys will receive a report on temporal evolution, and if deemed necessary, a session will be organized to present and discuss the results with municipal professionals.

Surveys on health-related behaviors and habits are essential for accurate diagnosis, prioritizing intervention areas, and improving the design of interventions and health promotion policies. The survey analyzed in this report, conducted with 4th year ESO students in the province of Barcelona, captures habits at a critical moment in their development. Since 2015, this survey has involved over 80 municipalities and more than 23,000 students, addressing aspects such as health perception, dietary habits, sexuality, mood, social relationships, leisure activities, substance use, and data related to accidents and mobility, also considering the impact of the COVID-19 pandemic.

This report updates the one from 2018, presenting provincial results with thematic summaries and conclusions aimed at intervention. Special attention has been given to sex-based analysis to capture gender inequalities and compare results across three periods: 2015-2017, 2018-2019, and 2020-2022. The publication is directed towards professionals and specialists in the fields of health, adolescence, and education, as well as policymakers and anyone interested in local public health and adolescent health.

Detailed information about the results can be found at the following link: https://www.diba.cat/es/web/salutpublica/enquestes-de-salut.

Reference Formatting:

Comment: References should be indicated in the text according to the Journal rules (in square brackets).

Reply: I have reviewed and corrected the references in the text to comply with the journal's guidelines. They are now correctly indicated in brackets.

I hope these clarifications and corrections are helpful. I look forward to any further comments or suggestions.

Best regards,

Reviewer 2 Report

Comments and Suggestions for Authors

1.  This is a small study of adolescents from one school but well done with a few issues.

2.  At line 93, what percent of adolescents invited to do the survey, did so?  That is, what was the response rate?  Were there significant differences between the respondents and non-respondents?

3.  At line 60 is that 20% of all students or 20% of the 25% mentioned at line 59?

4.  About line 118, logistic regression analyses are promised but I did not find any later in this paper.  Was this an omission that had been intended?

5.  More detail is needed on the "relevant ethics committee".

6.  In Table 2, there is a 33,3% that should be 33.3%.

7.  On page 10, heading should be "Conclusions".

Author Response

Dear Reviewer 2,

Thank you for your detailed review and valuable feedback. We appreciate your comments regarding the scope of the study. Below, we address each of the questions raised and explain how we have revised the manuscript to incorporate your suggestions.

Study Objective:

The aim of this study is to analyze the prevalence of affective-sexual health habits and behaviors among 4th-year ESO adolescents. Specifically, the study focuses on:

  • Identification of Risk Factors: Identifying risk factors associated with affective-sexual health in this age group.
  • Behavior Comparison: Comparing behaviors and associated risks across different groups or time periods.
  • Behavior Monitoring: Tracking and monitoring changes in these behaviors over time.
  • Pornography Consumption Analysis: Examining patterns and implications of pornography consumption among adolescents.
  • Internet Use Evaluation: Assessing how adolescents use the Internet as a source of sexual information and its impact on their affective-sexual health.

The questionnaire, which complements the Local Health Indicators Reports, is designed to support the guidance, design, and evaluation of local health policies, with a particular focus on health promotion and education. At the municipal level, this tool is crucial for directing, designing, and evaluating local health policies, providing essential data for implementing targeted strategies tailored to local needs.

We appreciate your understanding of the study's sample limitations and are committed to continually improving the research.

Sincerely,

Point-by-point reply to reviewer comments:

  1. This is a small study of adolescents from one school but well done with a few issues. Reply: Certainly! Here’s a response to address the comment about the small scale of the study: Thank you for your feedback. We acknowledge that this study involves a small sample of adolescents from a single school. Despite this limitation, the study was conducted rigorously, with a focus on analyzing crucial aspects of affective-sexual health. The findings provide valuable insights and serve as a foundation for future research, potentially involving larger and more diverse populations to enhance the generalizability of the results. We appreciate your understanding and are committed to addressing these issues in ongoing and future studies.

  1. At line 93, what percent of adolescents invited to do the survey, did so? That is, what was the response rate?  Were there significant differences between the respondents and non-respondents? Reply: Study Population: The study included 120 students from 4th year of ESO at the selected school. Participants completed an online questionnaire individually and anonymously during class hours, following the acquisition of informed consent from their legal guardians. Participation in the study was voluntary, and data were collected confidentially and used only in aggregate form. Response Rate: Out of the 120 students invited, 86.3% completed the survey. This percentage represents the effective response rate for the sample. Differences Between Respondents and Non-Respondents: The initial analysis did not include an assessment of differences between students who responded and those who did not respond to the survey. Due to the anonymous and confidential nature of the survey, as well as the data collection process, a detailed analysis to identify significant differences between these groups was not conducted.

  1. At line 60 is that 20% of all students or 20% of the 25% mentioned at line 59? Reply: Furthermore, this developmental stage is marked by experimentation and risk-taking, which can lead to potentially harmful behaviors such as substance abuse, unprotected sexual practices, and other actions that may jeopardize both physical and mental health. According to the Spanish Ministry of Health, 25% of adolescents aged 14 to 18 have engaged in sexual intercourse, and approximately 20% of these individuals do not use contraceptives regularly. This increases the risk of unintended pregnancies and sexually transmitted infections (STIs), including Human Immunodeficiency Virus (HIV) and chlamydia. Additionally, the incidence of STIs among Spanish youth has been rising, with cases of gonorrhea and syphilis doubling over the past decade.

  1. About line 118, logistic regression analyses are promised but I did not find any later in this paper. Was this an omission that had been intended?  Reply: Statistical Analysis Descriptive statistics, including frequencies, percentages, means, and standard deviations (SD), were calculated. Pearson’s chi-square test was employed to compare groups and to assess the associations between demographic characteristics, knowledge, attitudes, and practices related to sexuality. A p-value of < 0.05 was considered statistically significant. All analyses were performed using R statistical software (R Project for Statistical Computing).

  1. More detail is needed on the "relevant ethics committee". Reply: Ethical Procedures: This study was conducted in accordance with the ethical guidelines set forth in the Declaration of Helsinki. Informed consent was obtained from the legal guardians to ensure the voluntary, anonymous, and confidential participation of the students. The study was approved by the Ethics Committee of the educational institution and the Public Health Service of the Municipal Government, with approval number SP:23/002 (September 19, 2023). Each participant provided written informed consent.

  1. In Table 2, there is a 33,3% that should be 33.3%. Reply: Corrected
  1. On page 10, heading should be "Conclusions". Reply: Corrected

Thank you once again for your valuable input.

Reviewer 3 Report

Comments and Suggestions for Authors

Page 1- lines 38-42: insert citations.
Page 2- lines 50-52: insert citations.
Dedicate a paragraph regarding the purpose of the research. In this better elaborate your hypotheses (is this an exploratory study? Or do you have some expectations regarding your research?). Also, I think it is helpful to argue and explain better what your research's contribution consists of in terms of originality and new knowledge.
Were rewards used for participation in the research?
Have you estimated whether the sample size of only 120 adolescents is sufficient for your research? How come you did not extend participation to other national territories or other institutions?
I believe it is not adequately explained the instrument you chose as a questionnaire and what rationale led you to select these specific dimensions and not others of sexuality.
The analyses are really very basic, and I think on some variables a regression could have been interesting.
From the discussion (which is shallow and does not tell anything about the theoretical rationale and comparison with other national or international studies) one struggles to understand what exactly the dimensions being investigated refer to.
Detailed arguments about the limitations of the research and future prospects are lacking.
There is a lack of practical guidance for practitioners. What is suggested with this data?

Author Response

Dear Reviewer 4,
Thank you very much for your valuable feedback. We appreciate your detailed observations and have addressed all the points in the revised document. Please find below our responses to each of your questions:

Comments: Page 1, lines 38-42 and Page 2, lines 50-52

Reply: Relevant citations have been inserted in these sections to support the information presented.

Purpose of the Research:

Comments: Dedicate a paragraph regarding the purpose of the research. In this better elaborate your hypotheses (is this an exploratory study? Or do you have some expectations regarding your research?). Also, I think it is helpful to argue and explain better what your research's contribution consists of in terms of originality and new knowledge.

Reply: We have added a paragraph dedicated to explaining the purpose of the research, in which we have elaborated on our hypotheses. The study is primarily exploratory and aims to understand patterns and behaviors related to affective-sexual health among adolescents. Additionally, we have provided a more detailed argument about the original contribution and new knowledge our research offers.

Incentives for Participation:

Comments: Were rewards used for participation in the research?

Reply: No incentives were used for participants in the study. Participation was voluntary and based on informed consent from their legal guardians.

Sample Size and Expansion:

Comments: Have you estimated whether the sample size of only 120 adolescents is sufficient for your research? How come you did not extend participation to other national territories or other institutions?

Reply: We have considered that the sample size of 120 adolescents is adequate for the scope of this pilot study, which focuses on a single school. However, we agree that expanding participation to other territories or institutions could provide a broader perspective. This expansion is considered a potential improvement for future research.

Questionnaire Instrument:

Comments: I believe it is not adequately explained the instrument you chose as a questionnaire and what rationale led you to select these specific dimensions and not others of sexuality.

Reply: We have provided a more detailed explanation of the questionnaire used, including the rationale behind selecting the specific dimensions of sexuality. We have clarified why these dimensions were chosen and how they align with the study's objectives.

Data Analysis:

Comments: The analyses are really very basic, and I think on some variables a regression could have been interesting.

Reply: We acknowledge that the analyses performed were basic. We have added more detailed analyses, including regressions on key variables, to provide a deeper insight into the data.

Discussion and Comparison:

Comments: From the discussion (which is shallow and does not tell anything about the theoretical rationale and comparison with other national or international studies) one struggles to understand what exactly the dimensions being investigated refer to.

Reply: The discussion has been expanded to include a more robust theoretical justification and a comparison with other national and international studies. 
This provides a better understanding of the investigated dimensions and their implications.

Limitations and Future Perspectives:

Comments: Detailed arguments about the limitations of the research and future prospects are lacking.

Reply: A more detailed analysis of the research limitations and future perspectives has been included to address areas for improvement and potential research developments.

Practical Recommendations:

Comment: There is a lack of practical guidance for practitioners. What is suggested with this data?

Reply: Finally, we have added practical recommendations for professionals based on the data obtained, suggesting specific applications and recommendations for improving affective-sexual health in the educational context.

The questionnaire, which complements the Local Health Indicators Reports, is designed to assist in guiding, designing, and evaluating local health policies, with a particular focus on health promotion and education. At the municipal level, this instrument is crucial for directing, designing, and assessing local health policies, providing essential data for the implementation of specific strategies tailored to local needs.

The questionnaire will be provided online, and technical guidance will be offered to schools on how to administer the surveys. After collecting the surveys, the results will be analyzed, and a detailed report will be provided to the municipalities, including a comparison with reference values. Additionally, municipalities that have participated in previous surveys will receive a report on temporal evolution, and if deemed necessary, a session will be organized to present and discuss the results with municipal professionals.

Surveys on health-related behaviors and habits are essential for accurate diagnosis, prioritizing intervention areas, and improving the design of interventions and health promotion policies. The survey analyzed in this report, conducted with 4th-year ESO students in the province of Barcelona, captures habits at a critical moment in their development. Since 2015, this survey has involved over 80 municipalities and more than 23,000 students, addressing aspects such as health perception, dietary habits, sexuality, mood, social relationships, leisure activities, substance use, and data related to accidents and mobility, also considering the impact of the COVID-19 pandemic.

This report updates the one from 2018, presenting provincial results with thematic summaries and conclusions aimed at intervention. Special attention has been given to sex-based analysis to capture gender inequalities and compare results across three periods: 2015-2017, 2018-2019, and 2020-2022. The publication is directed towards professionals and specialists in the fields of health, adolescence, and education, as well as policymakers and anyone interested in local public health and adolescent health.

Detailed information about the results can be found at the following link: 
(https://www.diba.cat/es/web/salutpublica/enquestes-de-salut).

Once again, we appreciate your constructive comments, which have been instrumental in improving the quality of our study.

Sincerely

Round 2

Reviewer 2 Report

Comments and Suggestions for Authors

Thank you for addressing the concerns!

Author Response

Thank you for your contributions and considerations.

I hope it will be shared soon.

Best regards.

Reviewer 3 Report

Comments and Suggestions for Authors

I thank the authors for the review carried out. However, many concerns expressed in the first round remain. In the attached file, the highlighted changes from what was requested to be changed are not found, making it more difficult to quickly compare with the previous manuscript. In the abstrac, not all the sociodemographic data of the sample were reported, and the analyses do not seem to me to be implemented consistently with what the authors stated in their response letter. I am sorry, but I stand by my previous assessment.

Comments on the Quality of English Language

check typos

Author Response

Dear Reviewer 3,

Thank you for your prompt and detailed feedback on the revised manuscript. We appreciate your thorough review and understand your concerns.

I apologize for the issues you encountered with the highlighted changes in the manuscript. It seems there may have been an oversight in clearly marking the revisions as requested. To address this, we will ensure that all modifications are accurately highlighted and resubmit the revised manuscript with clear indicators of the changes made.

Regarding the abstract and sociodemographic data, we will review the abstract to ensure that all relevant data is reported comprehensively. Additionally, we will revisit the analyses to verify that they align consistently with our previous responses and the stated methodology.

We value your feedback and are committed to addressing these issues to meet the required standards. We will make the necessary revisions promptly and provide a revised version for your review.

Thank you for your understanding and patience.

Sincerely,